# Progesterone-Related Diabetes Mellitus in the Bitch: Current Knowledge, the Role of Pyometra, and Relevance in Practice

**DOI:** 10.3390/ani14060890

**Published:** 2024-03-14

**Authors:** Álan Gomes Pöppl, José Lucas Xavier Lopes, Taís Bock Nogueira, Denise Iparraguirre da Silva, Bruna dos Santos Machado

**Affiliations:** 1Department of Animal Medicine, Faculty of Veterinary, Universidade Federal do Rio Grande do Sul (UFRGS), Porto Alegre 91540-000, Brazil; 2Veterinary Sciences Post-Graduating Program (PPGCV), Universidade Federal do Rio Grande do Sul (UFRGS), Porto Alegre 91540-000, Brazil; lxaviier@hotmail.com (J.L.X.L.); taisbock2@gmail.com (T.B.N.); 3Veterinary Endocrinology and Metabolism Service (SEMV), Faculty of Veterinary, Universidade Federal do Rio Grande do Sul (UFRGS), Porto Alegre 91540-000, Brazil; deniipa@gmail.com (D.I.d.S.); brunadsm@ufrgs.br (B.d.S.M.)

**Keywords:** estrus cycle, diestrus, pregnancy, growth hormone, insulin resistance, insulin receptor, spaying, aglepristone, dog

## Abstract

**Simple Summary:**

Canine diabetes mellitus (CDM) is a multifactorial condition caused by insufficient insulin secretion, inadequate insulin action on peripheric tissues, or both. This scenario leads to increased blood glucose levels, which causes excessive urine output and thirst as the main clinical signs. Life-long insulin treatment is mandatory to control clinical signs and avoid life-threatening CDM complications. However, some female dogs may have a transient form of CDM associated with peripheral insulin resistance due to physiologically hormonal fluctuations’ impacts during their reproductive cycle. The resolution of antagonisms on insulin action may induce diabetes remission and allow insulin therapy interruption. This work aims to review the impacts of the reproductive physiology of the bitch on CDM risk and to discuss helpful steps to better manage entire diabetic bitches and eventually improve remission rates. Preventive measures are also discussed, as well as the role of pyometra, a common eventually life-threatening purulent infection of the uterus in entire females. Despite spaying being the best way to complimentarily treat diabetic female dogs, and to avoid reproductive-cycle-related CDM cases, the benefits and eventual harms should be considered before a decision individually.

**Abstract:**

Progesterone-related diabetes mellitus (PRDM) in dogs is known for its particular potential for diabetes remission. This narrative review aims to provide relevant detailed information on (1) the canine estrus cycle and its impact on canine diabetes mellitus (CDM) etiology and management, (2) the role of pyometra as a further cause of insulin resistance, and (3) useful individual therapeutic and preventive strategies. PRDM is recognized due to diestrus, exogenous progestogen exposure, pregnancy, and P4-production ovarian dysfunction. Pyometra represents additional inflammatory and septic negative influence on insulin sensitivity, and its diagnosis associated with CDM is therapeutically challenging. The estrus cycle’s hormone fluctuations seem to modulate peripheric insulin sensibility by influencing insulin receptor (IR) affinity and its binding capacity, as well as modulating tyrosine kinase activity. Pyometra was shown to negatively influence IR compensatory mechanisms to insulin resistance causing glucose intolerance. Spaying and pregnancy termination may cause diabetes remission in PRDM cases in a median time of 10 days (1–51). Pharmacological annulment of progesterone effects may benefit patients unable to undergo surgery; however, remission chances are virtually null. The ALIVE (Agreeing Language in Veterinary Endocrinology) project proposed new criteria for CDM diagnoses and subclinical diabetes recognition. These new concepts may increase the frequency of a PRDM diagnosis and, even more, its relevance. Spaying represents a preventive measure against pyometra and PRDM that should be individually assessed in light of its recognized benefits and harms.

## 1. Introduction

Diestrus is often associated with canine diabetes mellitus (CDM) [1,2], and this relationship has been recognized since the late 1950s [3,4]. Entire females and diestrus occurrence were identified as potential risk factors for CDM [5,6]. In regions where neutering is not a routine for the majority of the bitches, progesterone-related diabetes mellitus (PRDM) is probably the main form of CDM [6,7,8,9]. Less often, pregnancy may also trigger CDM [10,11]. Pyometra, another typical diestrus-associated disease [12], has also been related to CDM [4,11,13,14] and insulin resistance [15,16], not just because of the diestrus hormonal environment, but also due to the inflammatory and septic characteristics of this condition. These particular insulin-resistant CDM subtypes deserve attention since they are potentially reversible after removing the insulin-resistance cause [10,11,14]. Notwithstanding, molecular mechanisms involved in the insulin-resistant state of bitches in diestrus or with pyometra have been explored in the past years, bringing new insights about insulin resistance in dogs [16,17,18]. Both pyometra and CDM should be considered a differential diagnosis for polyuria and polydipsia in a bitch shortly after estrus, and both can be life-threatening diseases without proper treatment [3,4,13]. This is despite the concomitant occurrence of these conditions representing a darker prognosis and a therapeutic challenge; even so, diabetic remission can be achieved after suitable management of both conditions [14]. The complex interactions among reproductive hormones, pyometra, and diabetes mellitus in the bitch have been poorly explored and analyzed in scientific veterinary literature. In this way, this narrative review’s purpose is to summarize recent advances in the field of diestrus and pyometra-related insulin resistance and help clinicians understand why straightforward management is mandatory with intact diabetic bitches to improve their glycemic control and eventually achieve remission, as well as to inform potential preventive measures.

## 2. The Relevance of Progesterone-Related Diabetes Mellitus Subtype

The ALIVE project of the European Society of Veterinary Endocrinology defines diabetes mellitus as a heterogeneous group of diseases with multiple etiologies characterized by hyperglycemia resulting from inadequate insulin secretion, inadequate insulin action, or both [19]. The global prevalence of CDM from published literature ranges from 0.15 to 1.33% while sex and gonadal status (entire or desexed) predisposition to CDM varies according to the population studied [6,20,21,22,23,24,25,26,27]. However, it is generally accepted that entire females are at increased risk for CDM development due to the influence of diestrus [3,4,5,6,7,8,9,11,26]. Female dogs have a two-fold higher CDM incidence compared to males, and nearly 75% of the diagnosed dogs with DM are females [5,6,7,8,26,27]; however, this distribution can show a vast range. The reduction in synthetic progestin use as a contraceptive and systematic neutering of non-breeding dogs can make CDM incidence similar between males and females [28]. A study from Canarias found that 79.3% of CDM cases were in females, and 87% of them were entire [8]. A 95% female prevalence (19 out of 20 dogs) was documented in a case series in southern Brazil in the early 2000s, and 68.4% of those were in diestrus at the time of the CDM diagnosis [7]. In contrast, an American case series with 65 diabetic bitches found that only 8% were in diestrus at the time of the CDM diagnosis [29], and in some studies [20,23], desexed males appear to be at risk of CDM. 

The dog’s breed and, of course, its genetic background also play a role in the impact of the estrus cycle on insulin sensitivity and CDM predisposition. To note, in the Australian Terrier Dog, no significant difference in the effect of males and females on the CDM outcome was identified [30]. Nordic breeds like the Swedish and the Norwegian Elkhounds are at increased risk for progesterone-related diabetes mellitus (PRDM) [11]. However, in the American Eskimo Dog, an increased risk for CDM was documented in neutered females [31]. Genetic analyses have been characterizing dog breeds at risk of CDM, as well as breeds somehow protected [29,32,33], and despite certain haplotypes being overrepresented among susceptible breeds, to carry a genotype associated with the CDM phenotype does not warrant overt diabetes at a certain life stage, and complex environmental interactions are probably involved with disease development in genetically predisposed animals [2,32]. Heritability studies have suggested that the mode of inheritance of DM in certain breeds is polygenic, with no evidence for a single gene of a large effect causing CDM, and genetic uniformity for diabetes-susceptible genes would better explain individual susceptibility [30,31]. Notwithstanding, breed-related phenotypes and monogenic CDM forms could be better characterized soon [2].

Given the heterogeneous pathogenesis of CDM, common risk factors associated with insulin-resistant diabetes (IRD) such as obesity, diestrus, hypercortisolism, hypersomatotropism, progestins, and glucocorticoid exposure can act as triggers in previously susceptible dogs [2,5,6,9,11,28]. Environmental factors triggering types 1 and 2 DM in humans are well understood, and potentially other environmental factors such as diet, exercise, infections, and diseases (pancreatitis, pyometra) also have a role in these complex interactions regarding CDM pathogenesis [2].

The most important feature of PRDM cases is their potential reversibility after the removal of the P4 source, confirming the insulin-resistant nature of this condition [10,11,14]. When no remission is achieved after the end of diestrus, pregnancy termination, or ovariectomy and no other insulin resistance cause is identifiable, bitches are often reclassified as diagnosed with insulin-deficient diabetes (IDD) [28]. Despite some cases still having some positive staining cells for insulin in pancreatic islands and showing no evidence of immune-mediated insulitis, beta-cell mass and serum insulin are marked as reduced [8]. The resultant need for exogenous insulin therapy to control clinical signs and to avoid diabetic complications and death [29] after the resolution of insulin-resistance causes is often attributable to glucotoxicity [1,11,14], a phenomenon associated with beta-cell death after chronic exposure to hyperglycemia [34,35]. In contrast, ovariectomy in bitches was associated with increased insulin secretion capacity due to pancreatic islets’ hypertrophy [36]. Due to the complex influence of the estrus cycle hormonal fluctuations on insulin sensitivity and PRDM risk, understanding the particularities of the canine estrus cycle is mandatory to better manage and prevent those cases.

## 3. Canine Estrus Cycle: The Progesterone Perspective

The bitch is considered a monoestrus, typically non-seasonal species with polytocous spontaneous ovulation [37]. The cycle is divided into proestrus, estrus, diestrus, and anestrus. Inter-estrus intervals last around 6 (ranging from 5 to 12) months according to the breed, age, and environmental factors and correspond to the period between the end of estrus and the beginning of new proestrus [38,39]. The proestrus and estrus phases share similarities with the proliferative (follicular) phase of the menstrual cycle. They are marked by estrogenic influence, while diestrus is characterized by progesterone (P4) dominance similar to the secretory (luteal) phase of the menstrual cycle [37,38,39,40]. However, in the bitch, P4 starts rising during the proestrus and estrus before corpus luteum (CL) formation [37,38,39,40,41].

During proestrus, the bitch shows estrogen-related heat signs but refuses male mount [37,40]. Estrogens’ effects promote serial anatomical modifications in the reproductive tract, including fimbria proliferation, oviduct thickening, uterine horns’ elongation, and cervix enlargement [37,39,40]. Estrogens also cause vaginal epithelium and endometrial proliferation [42,43] and are associated with a P4-sensitizing effect by P4 receptor upregulation [44]. Estrogens positively influence theca cell steroidogenesis to avoid P4 secretion; at this point, it is just a hormonal precursor. Under luteinizing hormone (LH) influence, the theca cells produce androgens that will be further aromatized by the granulosa cells. The follicular stimulating hormone (FSH) not only promotes ovarian follicular recruitment and development but also upregulates aromatase activity within granulosa cells [38,45]. P4 levels can vary from <0.5 ng/mL at the beginning of the proestrus to >1 ng/mL at the proestrus’s end [40,41]. P4 rising concentration, associated with decreased estrogen production by follicular cells, determines estrus (heat) beginning and its characteristic behavior [37,39,46]. 

P4 levels keep rising during estrus and are around 2–3 ng/mL by the time of the LH peak and can range between 4 and 10 ng/mL by the time of ovulation [41,46]. Interestingly, P4 secretion during these phases derives from follicular cells (theca cells) before “luteinization” and may be seen as a kind of steroidogenic precursor leakage [37,39,45]. In contrast, during diestrus, P4 is the main steroidogenic product released from the CL due to diverse functional and morphologic modifications promoted by the LH on the follicular cells after ovulation [37,38,39,40]. 

Diestrus starts with the moment when the bitch starts to refuse the male mount [39]. After the CL formation, P4 concentration keeps rising during the first 2–3 weeks of the diestrus, reaching plateau values ranging from 15 to 90 ng/mL [37,39,46]. During this phase, the bitch is considered physiologically pregnant since there is no pregnancy recognition system in the dog. In this way, in non-pregnant bitches, CL remains functional for 10 to 30 days longer than observed in pregnant bitches, where CL suffers abrupt lysis by prostaglandin action around 65 days after fertilization as part of parturition onset [39]. Pituitary prolactin, as well as LH, are important luteotropic factors during diestrus, helping to sustain relatively high P4 levels [37,38,47]. P4 promotes and keeps endometrial growth as well as endometrial glandular activity and suppresses myometrial activity and intra-uterine leukocyte function while keeping the cervix closed [43,45]. These actions are fundamental for gestation maintenance but happen independently of fetal presence. The diestrus remains around 56–58 days in the pregnant bitch and can vary from 60 to 100 days in the non-pregnant ones. The end of diestrus is marked by P4 levels < 1 ng/mL and basal estrogen levels [37,38,39,40,41].

Anestrus is marked by the downregulation of ovarian activity and resultant basal levels of estrogens and P4 [37,41,46]. Uterine involution occurs, and also ovaries, vaginal epithelium, and mammary glands return to a basal state [39,45].

### Estrus Cycle Effects on Mammary Glands

The mammary glands also exhibit responses to the gonadal hormone fluctuations during the estrus cycle. While estrogens promote ductal and tubular growth, P4 initiates and promotes glandular development [39]. By the end of diestrus, a prolactin increase associated with a progressive reduction in P4 concentration can cause overt lactation and galactorrhea [38,45,47]. Some bitches would also show maternal behavior by adopting and protecting inanimate objects or other animal pseudocyesis [48]. Noteworthily, P4 also induces growth hormone (GH) production by ductal cells in the mammary glands [31]. This P4-induced GH should exert local autocrine and paracrine effects to promote mammary gland physiological proliferation and differentiation during diestrus. However, some mammary GH can be secreted into the blood and exert endocrine effects [49,50]. Endometrial hyperplastic changes [51], insulin resistance [52], and eventually overt acromegaly are some of the P4-induced GH effects [50,53]. Pituitary and gonadal hormone fluctuations during the canine estrus cycle are represented in Figure 1, as well as described GH variation during diestrus.

## 4. Estrus Cycle Effects on Insulin Sensitivity 

### 4.1. Progesterone: The Evil One?

The last proestrus days are characterized by discrete estrogen concentration reduction, while P4 starts to rise at the beginning of the estrus [37,38,39,40]. In this way, the estrogen/P4 ratio is marked as reduced during the late estrus period, and especially during diestrus. The PRDM is often observed during physiological diestrus but also may be secondary to pregnancy, ovarian remnant syndrome, and possible other ovarian diseases, or even iatrogenic due to the use of synthetic progestins [11,14,29,54]. Curiously, CDM secondary to P4-secreting corticoadrenal tumors is yet to be demonstrated in dogs [55]. Figure 2 depicts a chart of the progesterone-related conditions associated with PRDM in dogs. Despite the similarities between the dynamics of insulin resistance in pregnant bitches and women with gestational diabetes mellitus (GDM) [56], as well as several common clinical characteristics such as increased-severity near-full-term pregnancy, this concept is not well applied to bitches since the mechanisms behind GDM and PRDM are different [1].

Human GDM is characterized as carbohydrate intolerance (fasting glucose > 92 mg/dL or serum glucose > 180 mg/dL in one hour, or >153 mg/dL in two hours, after an oral 50 g glucose tolerance test) first identified during pregnancy, and these diagnostic criteria have often been judged over the years [57,58]. The ALIVE project has proposed new cutoffs for CDM, and briefly, diabetes can be assumed in a dog showing diabetes classic clinical signs (Pu/Pd, polyphagia, and weight loss) associated with random glycemia > 200 mg/dL, or eventually, in a dog showing persistent fasting glycemia > 126 mg/dL, but <200 mg/dL, associated with elevated glycated proteins such as fructosamine, independently of clinical sign presence [19]. Several comments and alerts accompany those criteria. This new view on diabetes diagnoses has created the figure of subclinical diabetes to describe patients that meet CDM diagnostic criteria; however, they do not have overt classical clinical signs. Most women with GDM do not develop clinical signs as well [57,58,59]. To date, no published research has applied these new concepts to investigate actual PRDM incidence in bitches during diestrus, pregnancy, or other P4-related conditions. However, a recent study found that only 3.5% of the dogs identified with random hyperglycemia between 126 and 200 mg/dL needed insulin treatment somewhere in the future [60], and none of those cases were in diestrus or pregnant at the time hyperglycemia was first documented. In women, GDM prevalence varies from 1 to 14% according to the studied population [57,58,59], and genetics [61], ethnicity [62], and overweight status [63] are known risk factors. The same relationships were also documented in the bitch, since certain breeds are at a clear increased risk for PRDM [11], and a previous overweight status was identified as an additional risk factor for PRDM in Elkhounds [64]. 

Insulin resistance is considered worse by the second half of gestation for both women [65] and bitches [56], and pregnant bitches are more insulin-resistant than bitches in diestrus [11,39,56,66,67]. After parturition, glucose tolerance is restored in about 90% of the GDM cases [1,68]; however, there is an increased risk for women who develop GDM to overt type 2 DM in the future. Also, after the first GDM episode, the condition is highly expected in subsequent gestations [68,69] especially in the presence of GAD antibodies [59]. In the bitch, better outcomes, including diabetes remission, were obtained after spaying surgery or medical pregnancy interruption [10,11], and in cases with spontaneous diabetes remission after the end of diestrus, relapse is expected in the next estrus cycle [3,4,10,14,29]. Nevertheless, it is not clear if bitches that achieve PRDM remission are at risk for future CDM by other causes [1]. Fetal macrosomia, maternal or newborn mortality, dystocia, and neonatal hypoglycemia are the main GDM complications in humans [66,69] and are suggested or weakly reported in dogs [10,11,66,67]. Despite it being natural to expect the same complications in diabetic pregnant bitches, there is very poor evidence in veterinary literature that diabetes during pregnancy causes the same risks for the bitch or the litter [1,10,66,67]; however, the pregnancy termination decision was common in pregnant diabetic bitches [10,14]. A small case series of gestational diabetes in dogs [10] found that dystocia occurred in 80% of the bitches reaching full-term pregnancy, and 50% were diagnosed as diabetic when seeking medical attention for dystocia. The study also reported a 27% neonatal mortality, which was considered higher than mortality rates described for puppies delivered naturally (10–15%) or by cesarean section (20%) [10]. Clinicians should be aware of these potential outcomes and the difficulties in regulating glucose levels during gestation [65,66,67]. 

The influence of diestrus on insulin sensitivity in bitches is often associated with P4-controlled GH overproduction; however, P4 itself seems to be guilty by its direct mechanisms. P4 was associated with reduced insulin binding to the insulin receptor (IR) [70] and suppressed intracellular insulin signaling pathways by multiple mechanisms [71] in adipocyte cultures. A more recent study in a mouse model [72] showed that P4 can cause increased hepatic glucose production via gluconeogenesis under the limited or impaired action of insulin, which may exacerbate hyperglycemia in diabetes where insulin action is limited. These effects cause insulin resistance and may predispose to glucose intolerance due to reduced glucose transport in target tissues in the bitch [73].

Glucose homeostasis biomarkers and their relationship with P4, GH, and insulin-like growth factor type-1 (IGF-1) were compared in PRDM-susceptible healthy Elkhounds during anestrus and diestrus and other breeds in anestrus and diestrus [74]. Elkhounds showed increased serum insulin, C-peptide concentration, and HOMA (homeostatic model assessment) for beta-cell function during diestrus compared with anestrus, while the HOMA for insulin sensitivity was lower, suggesting increased beta-cell function due to insulin resistance, while non-Elkhound dogs showed similar insulin secretion and sensitivity markers during anestrus and diestrus. GH and IGF-1 concentrations did not differ during diestrus in Elkhounds and non-Elkhound dogs when compared with anestrus results, but P4 did [74]. This finding suggests that P4-direct mechanisms can be primarily involved in insulin resistance in Elkhound dogs. However, each 1 ng/mL increase in GH serum concentration was associated with a 12.7% increase in the HOMA for insulin resistance (HOMA-IR), and IGF-1 concentration was positively correlated with C-peptide concentration in Swedish Elkhounds, showing that GH also exerts a strong insulin resistance effect during diestrus [75]. A study about natural estrous cycle effects in normal and diabetic bitches concerning glucose and insulin tests showed that intact naturally diabetic bitches in diestrus have severely suppressed insulin secretion compared with estrus or anestrus [76]. Another study [77] showed that P4 treatment caused mild insulin resistance together with a depletion of pancreatic insulin stores in the long run. 

In human GDM, not only P4 is involved, but also multifaceted mechanisms that probably involve hormonal, placental, genetic, and epigenetic contributions, as well as the activity level, diet/microbiome, and overweight/obesity [65]. Probably, most of these complex variables may also be involved in PRDM in dogs [2]. Regarding hormonal factors, cortisol, prolactin, and placental lactogen worsen P4-induced insulin resistance during pregnancy [70]. Hyperprolactinemia is a known cause of insulin resistance in humans, and despite prolactin’s rise during diestrus in dogs as a luteotropic factor [37,38,47], the role of prolactin corroborating insulin resistance induction in dogs is unknown. In contrast, mammary GH endocrine secretion in response to P4 does not occur in women [1].

### 4.2. How Does Growth Hormone Cause Insulin Resistance?

The diabetogenic role of P4-controlled GH overproduction in dogs has been well documented since the early 1980s after the recognition of the first acromegaly case in a bitch exposed to synthetic progestogens [78]. GH is a classic insulin counterregulatory hormone [79] and can reduce IR density in cell membranes and secondary glucose uptake in insulin-target tissues [80]. Other authors argue that IR downregulation is secondary to the hyperinsulinemia provoked by the GH [34]. The GH lipolytic effects also counteract insulin effects [79,80,81]. Accumulated evidence suggests that GH modulates insulin sensitivity by multiple mechanisms since GH and IGF-1 intracellular signaling pathways converge with insulin signaling pathways [52]. Chronic GH exposure is associated with reduced phosphorylation of the IR and the IR substrates IRS-1 and IRS-2, in muscle tissue, as well as reduced p85/IRS-1 association, and reduced PI3K activity. On the other hand, in the liver, excess GH is associated with the chronic activation of the pathway of IR/IRS-1/PI3K blocking activation by insulin. This aspect suggests that the liver is the main insulin-resistance site in response to GH. Moreover, insulin modulation action by GH also seems to involve mechanisms of signaling attenuation such as an increased expression of SOCS (suppressor of cytokine signaling) proteins and increased IRS-1 phosphorylation in serine residues [52,79]. Adiponectin and visfatin modulation by GH may also corroborate GH-induced insulin resistance [52]. 

Diestrus and medroxyprogesterone treatment were associated with absent GH suppression due to hyperglycemia in bitches [50,82] suggesting autonomous GH production, later shown arising from the mammary glands [49]. During diestrus, pulsatile pituitary GH secretion is perturbed by the mammary GH production, with a resultant increase in GH serum concentration and reduction in GH pulsatile secretion from the pituitary. During diestrus, P4 fluctuations positively correlate with GH serum concentration. In contrast, medroxyP4-treated bitches exhibit a chronic non-pulsatile increased serum mammary GH [50,82]. 

### 4.3. Do Estrogens Play a Role in Insulin Sensitivity?

Estrogens are poorly studied in veterinary medicine concerning insulin sensitivity impact. However, some entire diabetic bitches under insulin treatment may experience diabetes poorly controlled during proestrus. In this phase, there is a predominant estrogen effect. Anestrus’s estrogen basal levels (5–15 pg/mL) are progressively increased during proestrus due to follicular activity and can reach plateau levels above 60–70 pg/mL days before estrus [37,38,39,40]. Despite, in humans, some insulin-resistant states like the end of pregnancy or polycystic ovarian syndrome being associated with higher estrogen concentration [83], estrogens have been claimed to have serial protective effects in insulin-target tissues and other organs in post-menopause women [84]. These effects were documented in the pancreas (improved fasting insulinemia and glucose-stimulated insulin secretion), liver (modulatory effect on gluconeogenesis, improved insulin response, reduced hepatic insulin degradation), adipose tissue (improved insulin sensitivity, reduced oxidative stress), skeletal muscle (improved insulin-stimulated glucose absorption), heart (mitigation of insulin-induced cardiomyopathy and improved cardiac function), and vascular endothelium (increased nitric oxide production and vasodilation response) [84].

Sequential estrogen and P4 administration in healthy bitches in anestrus showed that the hypoglycemic effect of insulin was enhanced by estrogenization, together with insulin accumulation in Langerhans islets [77] despite some degree of glucose intolerance, and higher free fatty acids were also documented in bitches treated with estrogens [85]. 

An inhibitory estrogen effect on insulin-dependent glucose transporters (GLUT-4) was demonstrated in a mice model. Neutering reversibly increased GLUT-4 expression after estrogen repositioning in muscle and adipose tissue. The GLUT-4 upregulation after spaying leads to adipose hypertrophy and eventual adipose differentiation [86]; however, estradiol (E2) supplementation can restore weight, reduce appetite, and increase physical activity [87]. Notwithstanding, GLUT-4 cellular distribution in intraabdominal white adipose tissue was documented to be smaller in obese bitches compared with lean ones during anestrus [88]. Also, estrogen can negatively interact with an estrogen-responsive nuclear factor regulating lipoprotein lipase (LPL) expression [87], resulting in increased fatty acid influx to adipocytes after gonadectomy [89]. These findings may help to explain the weight gain after neutering in bitches [90]. Mechanisms behind the inhibitory effect of estrogen on GLUT-4 are not completely understood; however, estrogen-receptor (ER) subtypes are probably involved. ER-alfa stimulation activates GLUT-4 expression, and ER-beta stimulation, which is upregulated during the estrogenic phase of the estrus cycle, suppresses it [86]. Curiously, estrus was associated with worsened insulin receptor (IR) binding affinity in a study on insulin binding properties in bitches in different estrus cycle phases, but increased IR concentration seemed to compensate for it [18].

### 4.4. Estrus Cycle Effects on Markers of Insulin Sensitivity

The heterogeneous nature of PRDM in bitches can be realized through the different impacts of diestrus on markers of insulin sensitivity [74,75] and PRDM susceptibility among different breeds [11,26,31] as previously exposed. In this way, studies performed in more heterogeneous populations such as convenience hospital-population cases seen in clinical routines are less prone to identify a clear impact of the estrus cycle on insulin sensitivity markers [74]. The fasting glucose, serum fructosamine, fructosamine to albumin ratio, insulinemia, HOMA-B, HOMA-IR, insulinogenic index, and amended insulin to glucose ratio (AI: GR) were compared in a small heterogeneous breed (most being mongrels) group of “lean” (body condition scores 4–6/9) bitches in anestrus, estrus, or diestrus [15]. Age, weight, and BCS were similar among groups, and despite no statistically significant difference detected in the studied variables, the mean insulinogenic index [II = fasting insulin (μU/mL)/fasting glucose (mg/dL)] of bitches in diestrus was higher than the suggested cutoff for insulin resistance (>0.235) and 33% higher than II documented in bitches in anestrus or estrus [15]. 

Fasting insulin concentration, and its correlation with fasting glycemia explored by complex formulas such as HOMA-B and HOMA-IR, is not adequately validated for use in veterinary medicine since HOMA models were developed to be applied in humans [91], and HOMA models can fail to demonstrate insulin resistance in dogs [92]. The AI: GR corrected insulin/glucose ratio can be applied in the investigation of patients with hypoglycemia, even though its use for insulinoma diagnoses is questionable [93]. In these ways, fasting insulinemia and basal II are probably the simplest methods to assess insulin resistance in the dog in contrast with gold-standard clamp methods [94]. Nevertheless, a 40% reduction in insulin sensitivity in bitches in diestrus shown by the euglycemic–hyperinsulinemic glucose clamp technique was not associated with differences in baseline insulin or baseline glucose plasma concentrations [95].

Insulin resistance may induce hyperlipidemia [96]. No significative differences were found in total lipids, serum triglycerides, or total cholesterol in bitches during anestrus, estrus, or diestrus; however, intact diabetic bitches showed increased serum lipid results compared to non-diabetic ones, and results were higher during estrogenic and luteal phases compared with anestrus. Also, the triglyceride response to an insulin tolerance test (ITT) was greater in bitches in anestrus compared to bitches in seasons [97].

An intravenous-glucose tolerance test (IVGTT) was applied to assess insulin resistance in bitches in anestrus, estrus, or diestrus, but no differences in basal glycemia or insulinemia, neither glucose nor insulin responses, in the times studied could be identified [18]. The same bitches were submitted to ovariohysterectomy after IVGTT’s end and rectus abdominis muscle samples were collected for further studies. Muscle tissue membranes were extracted and submitted to tyrosine-kinase [17] and insulin binding experiments [18]. Those studies’ results brought evidence that the estrus cycle impacts insulin action at the IR and post-receptor steps.

The IR belongs to a subfamily of tyrosine kinase (TK) receptors, and so do the type I insulin-like growth factor (IGF-IR) and the orphan insulin receptor-related receptor (IRR) [98]. Insulin and IGF-1 may cross-bind their receptors with different affinities [52,79]. Rectus abdominis muscle membranes’ TK activity (CPM/µg of protein) was significantly (*p* < 0.001) smaller in estrus (57%) and diestrus (51%) in comparison with bitches in anestrus [17], suggesting a modulatory effect of the estrus cycle’s seasons in the TK-family receptor’s ability to phosphorylate intracellular substrates. Post-binding steps in insulin signaling, such as reduced TK activity, a decreased expression of insulin receptor substrate-1 (IRS-1), and increased levels of the p85α subunit of PI 3-kinase, are mainly involved as causes of insulin resistance in women with gestational diabetes mellitus [99]. 

IRs are a transmembrane tetrameric protein that consists of two α- and two β-subunits. Insulin binding to the extracellular monomeric α-subunits leads to the intracellular activation of the intrinsic kinase activity of the β-subunits [81,98]. Two ligand binding sites are often recognized in each α-subunit monomer due to their curvilinear Scatchard plots, and negative cooperativity: the low-affinity/high-capacity (S1) and the high-affinity/low-capacity (S2) binding sites [18,100,101]. The insulin–IR interaction is reversible, and at equilibrium, for each insulin–IR complex formed, another insulin–IR complex dissociates at the same rate. The dissociation constant (*K*d) is then considered to be the inverse association constant (*K*a) and depicts the insulin-free concentration needed to saturate half of the IR in the system [102], or in competitive binding experiments, the cold insulin concentration needed to reduce maximal ^125^I-insulin binding by 50% [18,100,101,102]. Therefore, *K*d is assumed as a measure of tissue insulin resistance, and the higher the *K*d, the smaller the insulin sensitivity. 

Insulin binding studies in the rectus abdominis muscle of bitches showed *K*d values with a 4.3-fold and 2.8-fold increase in estrus and diestrus, respectively, in comparison with *K*d values of bitches in anestrus (*p* < 0.001) [18]. However, this receptor-level resistance was accompanied by a 2.4-fold and 3.5-fold increase in the maximum binding capacity (*BMax*) of the muscle tissue’s membrane of bitches in estrus and diestrus, respectively, compared with anestrus. These results highlighted that muscle tissue became more insulin-resistant during estrus and diestrus due to elevated *K*d values. However, the maximum insulin binding capacity during these estrous cycle phases has increased and seems to compensate for the lower binding affinity, resulting in absent differences in basal fasting glycemia and insulinemia, as well as insulin and glucose responses during an IVGTT [16,18]. Curiously, different physiological conditions seem to modulate insulin binding characteristics only at the high-affinity/low-capacity binding sites in dogs [18,101], suggesting that the low-affinity/high-capacity binding sites could have a more constitutive role, less susceptible to modulation. Chronic exposure of bitches to estradiol and P4 caused insulin resistance in the whole body, but primarily in the skeletal muscle [103], and the estrogenic phase of the canine estrus cycle was associated with reduced peripheral (mainly muscle) insulin sensitivity [104]. Therefore, reduced muscle TK activity, as well as an altered modulation of insulin binding affinity and maximum capacity, may play a role in insulin-resistance induction and IRD predisposition in dogs [17,18]. 

## 5. Pyometra and Its Impact on Insulin Sensitivity

Pyometra is a common illness in middle-aged to older intact female dogs and is characterized by an acute or chronic suppurative bacterial infection of the uterus post-estrus with the accumulation of inflammatory exudate in the uterine lumen, leading to several local and systemic clinical signs [12,105,106]. The condition is considered P4-dependent since P4 effects within the uterus help the infection development, which can be life-threatening. Most cases develop during diestrus, and clinical manifestation often occurs within four months of estrus [106,107,108]. Pyometra is commonly associated with previous cystic endometrial hyperplasia (CEH), despite that the condition can occur independently [12,108]. Mammary GH overproduction due to P4 seems to also have a role in the pathogeny since IGF-1 is potentially involved in endometrial hyperplasia during diestrus, predisposing to the CEH–pyometra complex (CEH–P) [51]. 

Since CDM and pyometra were first correlated back in the 1960s [13], very few published studies explored pyometra as a cause of insulin resistance [11,14]. Pyometra represents an additional insulin resistance factor over diestrus due to its inflammatory and septic nature [14,105,109]. Similar to diestrus and pregnancy, CEH–P can overt DM in some bitches [11], and ovariohysterectomy may cause DM remission [14]. The former study that proposed a link between CDM and pyometra suggested that both conditions were preceded by obesity, suggesting the existence of a diabetes-mellitus–obesity–pyometra syndrome [13]. At that time, most CDM case presentations were considered closed-cervix pyometra cases due to obesity, polyuria/polydipsia, and a recent estrus history [4,13]. Pyometra is considered more prevalent among certain breeds, reinforcing a potential genetic role in pyometra susceptibility [12], and some were also overrepresented among dogs with CDM [13,32,33,39]. Despite pyometra being documented in 17% of a population of female Elkhounds with PRDM [11], the Swedish Elkhounds have a relatively lower risk of pyometra compared with other breeds but are highly susceptible to PRDM. Table 1 summarizes dog breeds with increased risk for pyometra and/or diabetes mellitus, while Table 2 summarizes the dog breeds with reported reduced risk for pyometra, CDM, or both. 

Insulin resistance secondary to pyometra was associated with higher basal serum insulin and higher area under the curve after an intravenous glucose load, as well as glucose intolerance during the IVGTT. These features normalized after pyometra surgical resolution and antibiotic treatment [16]. Reversal of the diestrus hormonal environment by ovariohysterectomy, as well as the removal of the infection focus, were considered the main mechanisms driving insulin resistance control. However, an eventual role of the antibiotic treatment modulating the microbiome and further impacting insulin sensitivity could also be involved [6]. Furthermore, altered insulin binding characteristics and reduced basal membrane TK activity were characterized in the muscle tissue of bitches with pyometra, suggesting that the mechanisms of pyometra-induced insulin resistance involve insulin signaling at receptor and post-receptor levels [16]. 

The mean *K*d values for the high-affinity/low-capacity insulin binding sites of bitches with pyometra were 6.27-fold higher than those observed in bitches in anestrus, and 2.63-fold higher than those in bitches in diestrus. Despite higher *K*d values, they were compensated for by increased *Bmax* capacity during diestrus, and there were no differences in this parameter in bitches with pyometra compared with diestrus. Failure to further increase the binding capacity due to the reduced IR binding affinity was associated with the insulin resistance and glucose intolerance documented in the IVGTT. Also, muscle membrane TK activity was reduced (65%, *p* < 0.001) in bitches with pyometra compared to anestrus but did not differ from TK activity documented in bitches in diestrus [16].

An increased concentration of inflammatory cytokines (TNF-a, IL-1, IL-6) under sepsis, or chronic inflammation conditions, inhibits insulin signaling through different mechanisms, including reduced TK activity and reduced insulin binding affinity [109,112,113,114,115]. In this scenario, the role played by the inflammatory environment in inducing additional insulin resistance seemed to explain worsened insulin sensitivity due to pyometra compared with the isolated diestrus influences [16]. Chronic inflammation due to obesity [116,117,118,119] or periodontitis [120,121], leading to insulin resistance, represents other examples of the negative effect of inflammation on insulin sensibility in dogs. These findings allowed the assumption that conditions such as diestrus and pyometra can exert insulin resistance at the IR and post-IR levels, and mechanisms such as IR upregulation represent a potential peripheral compensatory modulation [52,79,81]. Failure to counter-regulate muscle insulin sensitivity by increasing *Bmax* capacity was associated with glucose intolerance and higher basal insulinemia [16].

Hyperlipidemia could be caused by the chronic inflammatory status independent of inflammation-induced insulin resistance [114,115]. Pyometra-induced insulin resistance was not associated with an increased serum concentration of triglycerides and total cholesterol probably because of the brief period of insulin resistance before the diagnosis [18]. However, hypercholesterolemia was reported in 74% of the bitches with pyometra [12]. Serum glycemia abnormalities were reported in a very low percentage of bitches with pyometra, being hypoglycemia due to sepsis in 6%, and hyperglycemia in only 4% [12]. Despite the absence of fasting glycemia differences among bitches with pyometra or other estrus cycle phases [15], the severe increase in fasting insulinemia during pyometra allowed a clear demonstration of the insulin resistance status, employing insulin sensitivity indexes such as the II and AI: GR (both ~2-fold higher compared to diestrus), as well as HOMA-B and HOMA-R (both ~3-fold higher compared to diestrus). The same study showed a moderate Pearson’s correlation between the leukogram and II (r = 0.649, *p* = 0.012) and between the leukogram and HOMA-R (r = 0.709, *p* < 0.01) [15]. Fructosamine fails to evidence insulin resistance in bitches with pyometra, and showed reduced levels compared with bitches in diestrus [15,16], a finding considered to be the result of the lower albumin serum concentration levels [122] observed in the bitches with pyometra. Assessment of the fructosamine to albumin ratio to correct fructosamine misrepresentation was not shown to be helpful [16]. Serum fructosamine was previously considered a poor indicator of insulin resistance during diestrus [74], and reduced serum albumin due to inflammatory-mediated catabolism [112,113,114] that was documented in nearly a third of the bitches with pyometra [12] turns it even worse. 

Also, it is important to highlight that pyometra is more common in large-breed older bitches [12,13,106,107,108], and it is the mammary P4-controlled GH overproduction and the GH-related insulin resistance [78,79,80,81,82]. These pyometra epidemiological characteristics were associated with a significantly higher age and higher size of bitches in the studies that have compared insulin sensitivity among bitches with pyometra and other estrus cycle phases [15,16]. Since neither P4 nor GH were measured in these studies, the real cause of insulin resistance (older age, P4, GH, and/or inflammation) cannot be accurately assumed. However, the magnitude of the GH response to P4 as well as the tissue response to GH, and the glycemic/insulinemic responses to this phenomenon, are quite variable from dog to dog [49,50,74] and it seems reasonable for a pyometra-related septic and inflammatory environment to play an additional role in inducing insulin resistance in bitches during diestrus. Figure 3 shows a hypothetical causal diagram correlating factors potentially involved with PRDM in dogs.

## 6. How to Best Manage Progesterone-Related Diabetes Mellitus?

### 6.1. Initial Patient Evaluation

An often-asked question by the owners of diabetic pets is “Why did my pet become diabetic?”, and due to the heterogeneity of the CDM, the answer is usually “we don’t know” [123]. Notwithstanding, the main characteristic of IRD cases is the potential for remission after the identification and resolution of the target-tissue disorder causing insulin resistance [1,19]. In this way, properly identifying a possible diestrus role in the diabetes pathogeny of a bitch recently diagnosed as diabetic is central to better managing the patient.

Ideally, practitioners should run a checklist, looking for all possible factors influencing the diabetes origin, and that will potentially also impact treatment success [19,124]. Regarding PRDM, throughout reproductive history, anamnesis should be performed, including confidence regarding the status of neutered/entire (is the bitch confirmedly spayed?), time of spaying and eventual heat signs despite being spayed (ovarian remnant syndrome?), regularity of the estrus cycle (ovarian diseases or another chronic systemic disease such as hypothyroidism or hypercortisolism?), heat signs in the preceding two months (diestrus influence?), contraceptive methods (synthetic progestin exposure?), mating history (can the bitch be pregnant?), and finally clinical signs of systemic disease (hyporexia/anorexia, lethargy/depression, vomiting) that should alert toward possible diabetic ketoacidosis; however, they can be due to pyometra or other systemic illness (pancreatitis, renal failure). 

Clinical evaluation can eventually reveal acromegalic features such as increased skin folding and increased interdental space, macroglossia, and respiratory stridor, as well as mammary masses (Figure 4 and Figure 5) [53,78]. GH overproduction was also demonstrated in neoplastic mammary glands, and after malignant transformation, GH production by altered mammary tissue can turn P4-independent [75,125,126]. The identification of mammary masses should also raise suspicion about an insulin-resistant status.

Ideally, a minimum database including a complete blood count, a complete serum biochemistry profile including ketones and electrolytes, a blood gas analysis, a urine analysis (physical, chemical, and sediment), and abdominal ultrasound should be recommended to every newly diabetic dog [39,124]. In the bitch, these complementary exams will be helpful to assess eventual diabetic complications (i.e., dyslipidemia, increased liver enzymes, pre-renal azotemia, electrolyte abnormalities, ketoacidosis, subclinical bacteriuria) as well as pieces of evidence of potential causes for the diabetic state (i.e., Cushing’s syndrome, pancreatitis, pyometra, and others). Abdominal ultrasound can help identify uterine (wall hyperplasia, cystic hyperplasia, fluid collection) and ovarian (cysts, corpus luteum, neoplasia) abnormalities, as well as eventual ovarian remnant tissue or uterine stump abnormalities. Abdominal ultrasound may also show evidence of cystitis and pyelonephritis when present. Even though glycosuria is recognized as a risk factor for subclinical bacteriuria [29], it is relevant to state that the actual International Society for Companion Animals Infectious Diseases (ISCAID) recommendations do not recommend urine cultures in the absence of urinary tract infection clinical signs due to the potential for multi-resistant bacteria selection and the lack of evidence of harm due to not treating subclinical bacteriuria [127].

Insulin and P4 serum concentrations are also useful in the initial screening of the entire diabetic bitch. Despite most diabetic dogs showing basal insulin levels at the diagnosis, this finding is not pathognomonic of IDD, and can be secondary to initial glucotoxicity in IRD cases [8,29]. On the other hand, finding insulin levels within the reference range or above in a patient with diabetes indicates insulin resistance and an exciting potential for diabetic remission after removing the factor(s) causing insulin resistance. Serum P4 concentration can easily support a diestrus diagnosis, as well as vaginal cytology, and both are useful tools for estrus cycle phase monitoring [37,41,42,128,129]. Spayed bitches should have typical anestrus cytology with few basal cells. Diestrus cytology is marked by increased cellular content including parabasal and intermediary cells, neutrophils, and the eventual presence of the “metestrus cells” shortly after the end of estrus [39,40,41,42]. Canine GH assays are not commercially available worldwide, while there are assays to measure IGF-1 as a marker of the GH influence. However, to date, there is no cutoff for IGF-1 measurement in dogs to assume a negative influence of GH on insulin sensitivity, and IGF-1 concentration can overlap among insulin-resistant or insulin-sensitive bitches during diestrus [74]. In contrast, IGF-1 levels may be used to indirectly monitor diestrus influence in GH [130].

### 6.2. Treatment Goals and Recommendations

The ALIVE project has proposed treatment goals and physiological mechanisms through which these aims are achieved [19]. To date, PRDM cases should be treated the same way, and the resolution of the P4 and/or inflammatory-related conditions is crucial. Despite that spaying or pregnancy termination as a sole treatment without chronic insulin treatment was previously shown to induce remission [10,11], undergoing general anesthesia in a dehydrated uncompensated diabetic dog represents additional risks to a generally uncomplicated procedure [131]. Moreover, insulin treatment was associated with better outcomes in PRDM bitches treated through spaying and was considered crucial to protect beta cells from undergoing glucotoxicity while the procedure is programmed and while the insulin resistance decreases after P4 and inflammatory-related condition resolution [10,11,14,128,129]. Insulin treatment is also important to avoid cataracts and diabetic ketoacidosis (DKA) development [29,124].

In this sense, the initial long-term insulin regimen adopted can follow the same standards regardless of a PRDM diagnosis. Low twice-daily doses of intermediated or long-acting insulin suspensions or insulin analog solutions can be applied to try to cover the patient’s basal and post-prandial insulin requirements, taking into account local regulatory rules, costs, patient’s size, and owner‘s issues, for example [29,124,132]. Alternatively, a basal–bolus insulin treatment system can be proposed for particular cases, trying to better cover basal and post-prandial insulin requirements [132]. The insulin dose to stabilize a patient with diabetes is assumed as the difference between the patient’s needs and the patient’s insulin production [9]. In this way, it is difficult to assume a general standardized suggestion regarding the initial insulin dose in IRD cases since the patient’s insulin-resistance degree (i.e., presence of interferences of insulin action on target cells) and residual beta-cell secretory capacity are unknown. Starting doses of insulin recommended in actual guidelines can be used according to the veterinarian’s description [29,124,132,133]. Dietary management of patients with diabetes is also highly advisable and has serial benefits for the patient [29,124]. Not only is the diet content of macronutrients such as protein, fats, fiber, and simple carbohydrates important [134] but also the starch source is associated with better glycemic [135] and lipemic outcomes [136]. 

Serial insulin dose adjustments can be suggested following monitoring guidelines, considering the control of the clinical signs, preferably using standardized scoring systems such as the Diabetes Clinical Score and glycemic variability evaluated in the blood, interstitium, and/or urine [19,29,124]. Despite good clinical control being reached in dogs, maintaining daily mean glycemic values between 80 and 250 mg/dL [29], there is no prospective high-level evidence that setting a specific glycemic range could be correlated with a specific treatment outcome, including remission [19]. Fructosamine values in diabetic monitoring have more value when compared within the same patient over time, using the same laboratory, and in light of the clinical picture. To date, there are no validated fructosamine intervals to determine poor or good glycemic control. Is also important to note that fructosamine levels are affected by factors such as lipemia, plasmatic protein concentration, and turnover rate. The veterinary endocrine societies from the USA (Society for Comparative Endocrinology—SCE), Europe (European Society of Veterinary Endocrinology—ESVE), and Brazil (Brazilian Association of Veterinary Endocrinology—ABEV) endorsed the comments above and others on fructosamine serum measurement and they are available online [137].

### 6.3. When Is Surgery Indicated?

It is important to emphasize that achieving diabetic clinical or glycemic control in an intact bitch under P4 and/or GH influence due to diestrus, pregnancy, progestin administration, or an eventual pyometra can be very frustrating and unsuccessful [10,11,29]. By the way, waiting for “perfect” diabetic clinical control achievement before proposing surgery can delay and reduce the chances of diabetic remission [14,127]. In cats, diabetes mellitus secondary to acromegaly has incredibly high remission rates after successful hypophysectomy due to quick removal of the GH source [138,139] and the same results are not achieved with medical therapy [140,141]. Despite distinct species, the removal of the stimuli for the mammary GH overproduction in bitches with PRDM (spaying), as well as the removal of mammary masses eventually producing GH, is important to improve diabetic remission rates in dogs.

A diabetes mellitus diagnosis exerts heavy concerns in many owners, eventually leading to a decision of euthanasia [3,4,7,142,143,144]. In this way, the possibility of achieving remission is very welcome by the owners, and nearly 10% of the bitches spayed after the DM diagnosis achieved remission [14]. Another study achieved 58% remission after pregnancy termination in gestational diabetes in bitches [10]. As proposed for the surgical treatment of pyometra [12], the author’s opinion and experiences regarding PRDM are that spaying surgery should be performed as soon as the patient is hemodynamically stable to increase remission rates (Figure 6). Detailed considerations for anesthesia of patients with diabetes are available elsewhere [131]. Even patients presenting with DKA can undergo surgery after becoming stable [14,131,145]. In a study reporting diabetic remission in bitches after the resolution of P4 and inflammatory-related condition resolution, one case that presented with possible DKA (hyporexia, nausea, severe hyperglycemia, and marked ketonuria) due to ovarian remnant syndrome was submitted to remnant tissue removal surgery shortly after intensive care under fluid, electrolytes, and short-acting regular insulin therapy, and achieved remission four days later [14]. Many other cases achieved successful remission after this more aggressive approach regarding PRDM since then in our experience. 

Spaying should not be viewed as an elective procedure in these cases but as an urgency. Ovariectomy of bitches in the mid-luteal phase stops P4-induced GH release from the mammary gland, resulting in lower plasma GH levels, recurrence of GH pulsatility, and lower circulating IGF-I levels [130]. Theoretically, the sooner the surgery, the greater the remission chance [14,29]. However, diabetic remission in a bitch was already reported even after being spayed almost 12 weeks after the diabetes diagnosis [14]. Spaying induced diabetic remission for four months in a case report of a poorly responsive-to-insulin-treatment Schnauzer dog with concomitant hypercortisolism and multiple mammary masses [129]. Ovariohysterectomy was first recommended as a drastic therapeutic measure to avoid rapid clinical deterioration in the 1960s [4]. Advances in veterinary anesthesiology can make this procedure quite safer nowadays, rather than not free of complications [131]. In this way, bitches diagnosed with PRDM could benefit from intensive treatment under hospitalization to quickly restore fluid and electrolyte abnormalities identified while the glycemic control is provided by short- or intermediate-acting insulin therapy to stabilize the patient for spaying surgery in a shorter time after the diagnosis (e.g., 1–2 days). 

### 6.4. What Are the Next Steps after Surgery?

Even when spaying surgery is performed soon after the diabetes diagnosis, it is important to properly manage the owner’s expectancy and clearly state that the surgery does not warrant diabetic remission, but surely will improve diabetic control, and luckily will be associated with remission [14,29]. Long-term insulin treatment would still be needed for a few days to weeks in cases achieving remission, and client education regarding diabetes management and glycemic home monitoring is mandatory [10,14,128]. An arbitrary recommendation for an insulin dose reduction (e.g., 20–30%) after surgery can be applied, assuming that an important source of insulin action antagonism was removed. The growing use of continuous glucose monitoring systems (CGMSs) and generated data delivery to the owner and the veterinarian (e.g., FreeStyle Libre and LibreLinkUp application) make it easy for hypoglycemic tendencies recognition and prompt insulin dose adjustments [146,147]. Glycemic parameters within the reference interval of non-diabetic dogs (i.e., glycemia between 70 and 120 mg/dL, absent glycosuria) can be assumed as a trend to diabetic remission or insulin overdosing and therefore imply the possibility of episodes of hypoglycemia [19]. Eventually, a dog undergoing diabetic remission after spaying can be presented due to a hypoglycemic crisis [14]. When diabetic remission is suspected due to the above-mentioned factors, progressive insulin dose reduction can be tested (e.g., 25–50% reduction every 3–5 days) while glycemic values and clinical score are being monitored [27,29,124]. In this scenario, maintenance of a long-acting “peakless” basal insulin analog once daily (e.g., glargine 300 U) before complete insulin withdrawal would be safer to reduce hypoglycemic risk while insulin resistance is reducing, and endogenous insulin secretion is recovering [148]. Alternatively, prompt insulin interruption and checking for overt hyperglycemia or relapse in clinical signs in the following days would also be indicated [29]. The author prefers the second strategy when there is normal or increased endogenous insulin secretion by the time of diagnosis. 

Diabetic remission can be assumed when a dog previously diagnosed with DM using ALIVE criteria ceases to receive exogenous insulin therapy and shows no evidence of DM according to ALIVE criteria after 4 weeks [19]. The time to stop insulin treatment in a case series of gestational diabetes that achieved diabetic remission after pregnancy termination ranged from 7 to 21 days (median: 9) [10]. A case series of diabetic remission among Elkhounds reported a median time to insulin interruption of 11 days (range: 1 to 51 days). Another case series of diabetic remission after the resolution of inflammatory and P4-related conditions showed a range from 4 to 39 days (median: 10) to stop insulin after ovariohysterectomy [14]. Based on the findings of those studies and single-case reports [128,129], the estimated median time to stop insulin after surgery and progressively reduce the insulin dose when remission is achieved is around 10 days.

### 6.5. What to Do When Spaying Is Not an Option?

Spaying can be a sensible topic for some owners, and different points of view regarding this issue can arise during a consultation, influencing the decision to not undergo surgery. For example, a pregnant breeder dog diagnosed with diabetes near the end of gestation would be a potential case leading an owner to refuse pregnancy termination. Trying to help the patient with diabetes deliver her puppies can be particularly challenging [10] despite gratifying, and progressive insulin resistance and increased risk for DKA are expected closer to the full-term pregnancy with resultant increased risk for the mother and puppies, especially in older bitches [10,66,67]. Insulin doses and regimens can be reviewed according to the patient’s needs [27,29,124,132]. Perfect glucose control was quite unlikely in a study with gestational diabetes in dogs [10]; however, the patients were treated with insulin suspensions (NPH or intermediate porcine insulin), in a median dose of 0.65 U/kg q12h (range: 0.4–1.5 U/kg, q12h), and eventually more aggressive doses could have performed better. In contrast, hypoglycemia concern and time for judgment of the insulin-dose adjustments [29,124] could have delayed the achievement of effective insulin doses within this cohort before full-term pregnancy. Also, most bitches were diagnosed during the final third of pregnancy. Modern technologies in glucose monitoring and new insulin treatment protocols allow quicker and more aggressive insulin adjustments nowadays [124]. The insulin analog detemir could be a helpful option for patients poorly regulated with other insulin according to a recent study [149]; however, no entire female under diestrus was studied in this small case series. On the other hand, detemir insulin was useful in improving the management of a diabetic dog with concurrent Cushing’s syndrome and also prevented insulin-dependent treatment in canine patients with Cushing’s syndrome presenting with hyperglycemia [150]. Despite being quite different conditions, the insulin resistance pattern induced by the diestrus and glucocorticoid excess due to hypercortisolism was shown to be similar [95]. Regarding basal–bolus protocols, long-acting insulin analogs (e.g., glargine 300U, degludec) are the best options to employ as basal insulin, while intermediate-acting insulin such as NPH is considered the best option to employ as bolus insulin due to physiological insulin secretion patterns in a dog [132,151]. However, a basal–bolus protocol using NPH as basal insulin, and the short-acting analog lispro as bolus insulin, was helpful in improving glycemic control in a group of poorly regulated diabetic dogs [152]. 

Despite that the open cervix pyometra can be medically treated [12], concurrent diabetes mellitus should be assumed as a contraindication for medical treatment. Other times, the general patient condition or comorbidities (e.g., end-stage chronic kidney disease or heart failure), or the owner’s financial issues, can be impeditive to a surgical approach. In this scenario, increasing insulin doses, frequency, and eventual different insulin combinations can be applied to achieve ALIVE’s proposed treatment goals [19]. However, the owner’s and pet’s quality of life can be negatively impacted [142,143,153], and the risk of hypoglycemia increases in parallel with more aggressive insulin treatments [29,124]. So, trying to reduce P4-related insulin resistance can be attempted in other ways while waiting for the end of natural diestrus.

To date, pharmacological treatments applicable to PRDM cases are generally less effective compared with spaying. Aglepristone, a P4 receptor (PR) blocker approved for veterinary use, is typically used for parturition induction [154], pregnancy interruption [155], and pyometra medical treatment in bitches [12,106,108]. Two aglepristone doses (10 mg/kg, SC) 24 h apart were 100% effective in medically terminating mid-term pregnancy in bitches [156]. Its effects on the uterus allow cervix opening and uterine contraction inhibited by P4. The PR blockage can also revert P4 effects in the mammary glands despite having a small effect on estrus cycle length and a negligible effect on P4 concentration [157]. The aglepristone effect on mammary glands is effective in controlling local GH effects and reverts fibroadenomatous mammary hyperplasia in queens [158] and also reverts progestin-induced GH overproduction in dogs [159]. Aglepristone serial administration (10 mg/kg, SC) on days 1, 2, 9, and 17 after the diabetes diagnosis in entire bitches in diestrus induced reduction in GH concentration and better glycemic control, allowing insulin dose reduction after two weeks of the beginning of the treatment [158].

Trilostane, a 3β-hydroxysteroid dehydrogenase inhibitor commonly used in hypercortisolism treatment in dogs [160,161], also has the potential to decrease P4 secretion [155,162] and can eventually be applied in diabetic bitches during diestrus if aglepristone treatment is not available. Trilostane doses ranging from 2.2 to 6.7 mg/kg once daily for seven days cause a significant reduction in P4 concentration in pregnant bitches but are ineffective in inducing pregnancy termination [156]. Another study showed that trilostane given at a dose range of 3.3–5.3 mg/kg twice daily was able to significantly decrease P4 concentration during both the pituitary-dependent and the pituitary-independent parts of the luteal phase of healthy Beagle bitches [162]. However, neither study assessed trilostane impacts on GH or IGF-1 concentration, as well as an eventual impact on insulin sensitivity. Moreover, P4 levels in both studies remained compatible with diestrus levels, despite the P4-lowering effect of trilostane [156,162].

Dopaminergic drugs acting on the pituitary level could be considered with some potential to reduce P4-mediated insulin resistance by suppressing prolactin levels, such as cabergoline. Cabergoline administration at 5 mcg/kg daily was effective in reducing P4 levels below 1 ng/dL in 15 to 20 days; however, this treatment is associated with proestrus/estrus induction, and the effect on insulin sensitivity was not evaluated [163]. On the other hand, gonadotrophin-releasing hormone (GnRH) agonists such as a deslorelin 4.7 mg subcutaneous implant applied during diestrus successfully suppressed the estrus cycle in bitches since being applied every 4 months; however, a transient increase in serum P4 was documented before anestrus induction [164]. 

Sodium–glucose transporter type-2 (SGLT-2) inhibitors are being successfully applied in eligible feline IRD cases [165]. There is extremely limited information on SGLT-2 inhibitors in dogs [166,167]. Their possible role and benefits as adjunctive to insulin treatment [167] are to be shown and there are studies on this subject going on. Eventually, SGLT-2 associated with insulin therapy can represent an alternative treatment to avoid IRD-negative impacts on beta-cell function due to glucotoxicity in dogs. In contrast, euglycemic ketoacidosis and hypoglycemia are potential risks involved with this therapy to monitor [145,167]. 

### 6.6. Appliable Preventive Measures

Given all evidence toward the negative impact of diestrus and the P4-controlled GH overproduction on insulin sensitivity, and the generally increased risk for CDM among entire females, spaying can be claimed as the main way to prevent PRDM in the bitch. However, PRDM is more common among certain breeds [11], while others seem to be protected from DM when kept intact [31]. Spaying can be a protective measure not only against PRDM but also against mammary neoplasia development [168,169] and pyometra [170]. Since pyometra may additionally predispose a given bitch to PRDM, spaying preventive effects on both diseases are welcome. Knowledge about breed-related predisposition to both diseases as well as breed-related protection is important to better inform valuable preventive measures against pyometra and/or CDM (Table 1 and Table 2). Spaying is also important as a population control tool [171] and a helpful strategy for some behavior problems [172]. In contrast, it is also associated with several chronic complications including obesity [90,173], cognitive impairment [172], orthopedic diseases and certain types of cancer [174], and immune-mediated diseases [170], including eventually IDD due to immune-mediated mechanisms, leading to beta-cell loss [31], urinary incontinence [175], and even a suggested LH-dependent hypercortisolism variant [176]. Breed-specific recommendations regarding undergoing spaying surgery or not, as well as when to perform it, have been generated based on some of these aspects [174]. Due to these complex interactions between the impacts of sex hormones and the pituitary–gonadal axis imbalance after spaying [177], it is difficult to define the overall benefits and risks of spaying for a particular dog. The recognition of acromegalic features [78] in intact bitches should represent a stronger recommendation for spaying since the acromegalic phenotype can be considered a biomarker of the GH influence [178] (Figure 7). However, it is important to note that the acromegaly phenotype in dogs may rarely occur due to pituitary somatotrophs [179,180] or due to primary hypothyroidism [181,182,183]. Despite P4-controlled GH’s negative influence on insulin sensitivity being documented only during diestrus, continued intermittent GH exposure over the next estrus cycles can predict increased cumulative risk for DM, especially due to aging effects on beta-cell function [10]. Also, since mammary neoplasms can secrete GH [75,126], prompt surgical removal of identifiable mammary neoplasms is not only helpful for potentially reducing the odds of DM among entire bitches, but also may reduce potential negative impacts on the bitch’s lifespan and quality of life [184]. 

Many owners do not agree with spaying their dogs, and in these cases, other ways to try to prevent PRDM are welcome. Environmental risk factors for CDM have been explored in the literature and raised the “Nature or Nurture” dilemma in canine diabetology [73]. Several pieces of evidence obtained from retrospective studies point to factors such as obesity [5,6,23,24], diet [6,64,185], and glucocorticoid exposure [6,24] as relevant risk factors for CDM despite that the eventual underlying mechanisms are unknown. Within a diabetic Elkhound population composed exclusively of females with PRDM (10.4% being pregnant), previous obesity was shown as a relevant risk factor, as well as age [64]. In these ways, the knowledge regarding PRDM risk factors allows simple recommendations that can potentially reduce the risk of DM development in an entire female [123]. These may include rational use of glucocorticoids especially in older, breed-predisposed, and overweight entire bitches [6], feeding balanced commercial dog food as the main (>90%) daily source of energy and nutrients [5,7,64,90], and preventing overweight development based on promoting the owner’s education, adequate feeding recommendations, and regular physical activity [64,185,186]. Avoiding mating and the risk of pregnancy should also be a matter of concern since insulin resistance due to pregnancy can be more dangerous [10,11]. Eventually, the use of aglepristone to block P4 effects by the end of estrus [154] could also be considered as a preventive measure in highly susceptible bitches. Notwithstanding, the recent characterization of diabetes remission in a male dog suggested that much of the beta-cell biology is still to be unraveled in canine diabetology [1,2,187].

## 7. Conclusions

Far and away providing definitive recommendations and understanding of the features related to PRDM in the bitch, this review brings up relevant insights and reviews the main aspects involved in this CDM subtype of particular interest. The complex effects of P4 and P4-controlled growth hormone overexpression during diestrus or pregnancy represent a strong risk factor for DM development especially in older bitches. Genetics, pre-existing obesity, and nutritional aspects are potentially also involved; however, spaying may represent a preventive measure against CDM development for many bitches. Also, the risk for pyometra development is dramatically reduced in spayed bitches. The septic and inflammatory characteristics of pyometra additionally decrease insulin sensitivity during diestrus. Quickly spaying after the CDM diagnosis in a bitch in diestrus as well as prompt pyometra surgical resolution and pregnancy termination are associated with better outcomes and increased probability of diabetic remission. In this way, more aggressive and straightforward initial patient hydroelectrolytic stabilization and insulin treatment may allow quicker surgical management and potentially increase remission rates in bitches presenting with PRDM. Hypoglycemia monitoring should be carefully discussed with owners of recently spayed diabetic bitches to help with insulin adjustments and remission identification. Despite diabetic remission being a goal with a brilliant potential for improving the patient and owner’s quality of life, spaying may not be enough to promote remission, and the eventual negative impacts of gonadectomy should be discussed with the owners on an individual basis.

## Figures and Tables

**Figure 1 animals-14-00890-f001:**
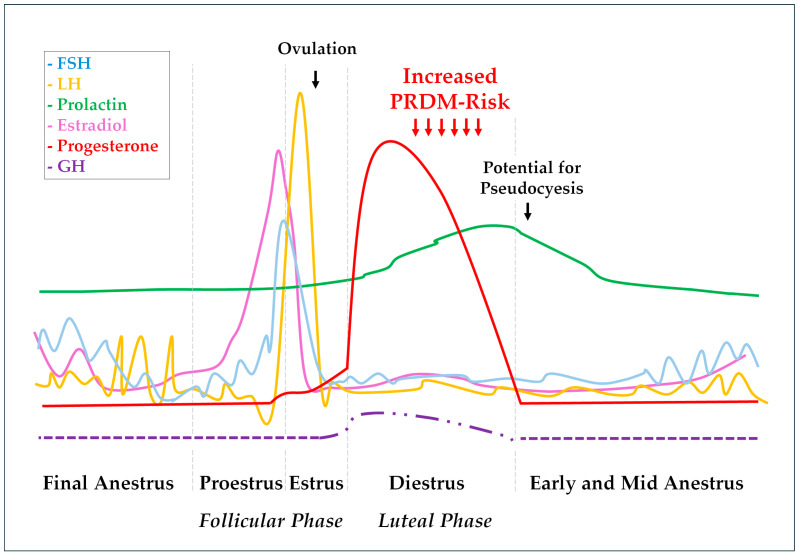
Hormonal fluctuations during the canine estrus cycle. The scheme represents pituitary gonadotrophin (FSH, LH, and prolactin) patterns, as well as resultant ovarian E2 and P4 responses. The P4 influence on basal GH concentration and pulsatility pattern are also represented. The FSH line (blue) represents fluctuations between 15 and 40 and 200 and 400 ng/mL during the estrus cycle, while the LH line (yellow) represents variations from 0.4 to 1.5 to 5 to 40 ng/mL, and the prolactin line (green) shows fluctuations ranging from 0.5 to 2 to 5 to 30 ng/mL. E2 (pink) and P4 (red) lines range from 5 to 10 pg/mL and 0.2 to 0.5 ng/mL when in basal levels to 45 to 120 pg/mL and 15 to 90 ng/mL, respectively [37]. Basal GH concentration during anestrus varies around 1.4 ± 0.2 µg^−1^ and can reach values around 2.3 ± 0.2 µg^−1^ 18–20 days after ovulation. During this early diestrus, the GH pulsatile secretion pattern is marked as reduced from ~5 to ~2 peaks/12 h as well as the pulse duration from ~41 to ~11 min compared with anestrus. All these GH pulse and concentration abnormalities were slowly mitigated during the subsequent weeks following P4 throwback to basal levels [47].

**Figure 2 animals-14-00890-f002:**
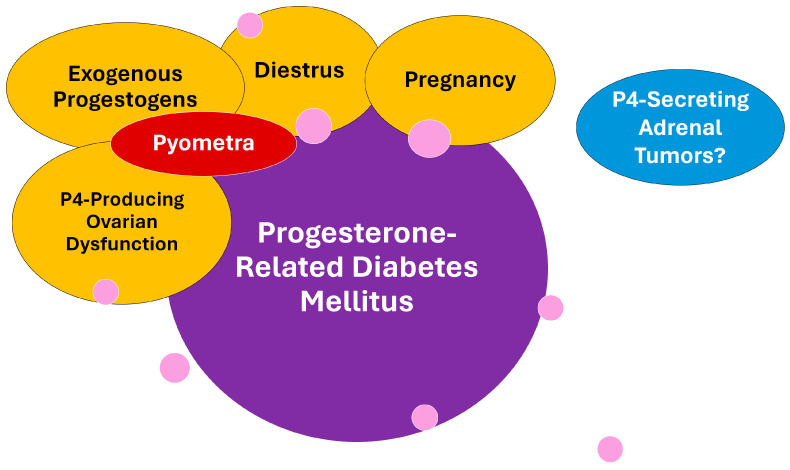
Conditions under progesterone influence that are involved with the progesterone-related diabetes mellitus (PRDM) subtype in dogs. Pyometra is a progesterone-dependent disease secondary to diestrus, exogenous progestogen exposure, or ovarian dysfunctions adding inflammatory and septic negative influence on insulin sensitivity and PRDM risk. To date, progesterone-secreting adrenal tumors were not associated with diabetes in dogs, despite being considered a common complication in cats. Pink circles represent general genetic and environmental factors that may influence PRDM occurrence and that are further detailed later in the text.

**Figure 3 animals-14-00890-f003:**
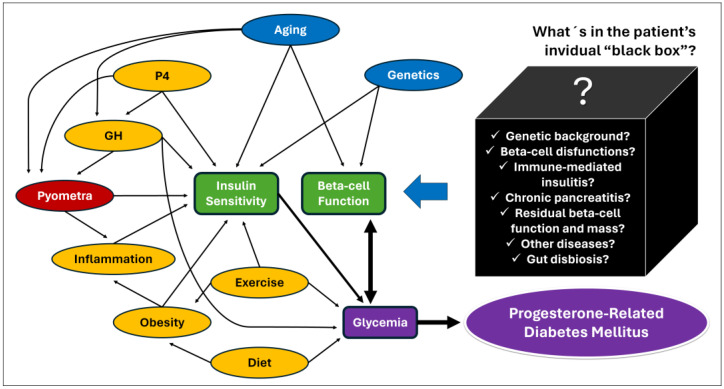
Hypothetical causal diagram for P4-related diabetes mellitus in dogs [2,5,64,123]. The arrows represent evidence obtained from studies with dogs, but also from studies with humans, animal models, and cell cultures. The main determinants of glycemia (purple) are beta-cell function and peripheric insulin sensitivity (red), both influenced by intrinsic factors such as age and genetics (blue). Factors shown in yellow can potentially be modified by proper patient management. Glycemia can also affect beta-cell function by glucotoxicity (heavy double arrow). Serial unknown factors may exert positive or negative effects on beta-cell function (“black box”) and influence the patient’s ability to increase insulin secretion to avoid hyperglycemia in an insulin-resistant environment. Those factors are potentially also determinants of diabetic remission after resolution of P4-related conditions.

**Figure 4 animals-14-00890-f004:**
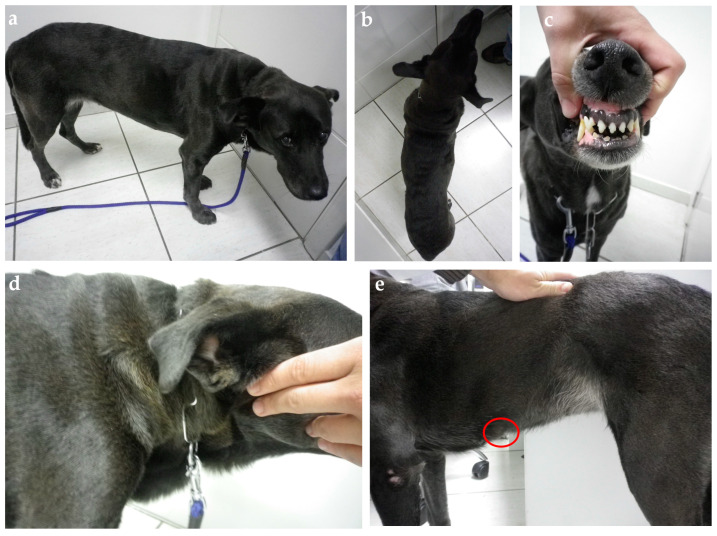
Ten-year-old mixed-breed dog with diabetes diagnosis three weeks after estrus. (**a**) The general patient’s appearance shows weight loss and increased thick folding of the skin over the withers (**b**) and neck (**d**). (**c**) Discrete widening of the interdental space and gingival hyperplasia. (**e**) A palpable flat mammary mass measuring nearly 3 cm in its highest diameter was identified during the physical exam (red circle). Insulin therapy was started (NPH, 0.4 U/kg, q12h) associated with a diabetic commercial food (Royal Canin Diabetic). On day 5, the insulin dose was adjusted to 0.55 U/kg, q12h, due to initial poor clinical and glycemic responses. Three days later, a hypoglycemic episode was documented. The insulin dose was reduced back to the initial dose, but new hypoglycemic episodes were documented in the following days. Insulin was interrupted on day 11 and diabetic remission was associated with natural P4 reduction in the second diestrus half. The owner refuses to proceed with spaying surgery, and the dog relapsed into diabetes five months later after the subsequent estrus. At this time, the bitch was spayed and underwent a mastectomy without achieving diabetic remission a second time. Her insulin need was dramatically reduced after surgery (0.2 U/kg, q12h). The dog was followed until death four years later and never achieved remission again despite maintaining a relatively “low” insulin requirement over time.

**Figure 5 animals-14-00890-f005:**
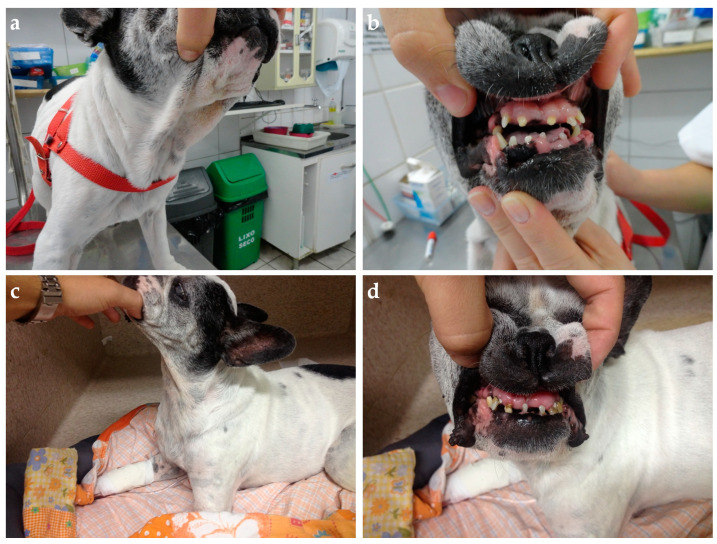
Nine-year-old Frech Bulldog presented with poorly controlled diabetes diagnosed after the last estrus five weeks before. (**a**) Thick skin folds in the neck and (**b**) marked widened interdental space with gingival hyperplasia and discrete enlarged tongue. The bitch also had a mammary mass and was submitted to ovariohysterectomy and mastectomy. Diabetic remission was achieved four weeks later. At fourteen years, the dog relapsed due to chronic kidney disease. (**c**) Note the clear reduction in soft tissue overgrowth and skin folding; (**d**) however, bony changes documented in the mouth persisted.

**Figure 6 animals-14-00890-f006:**
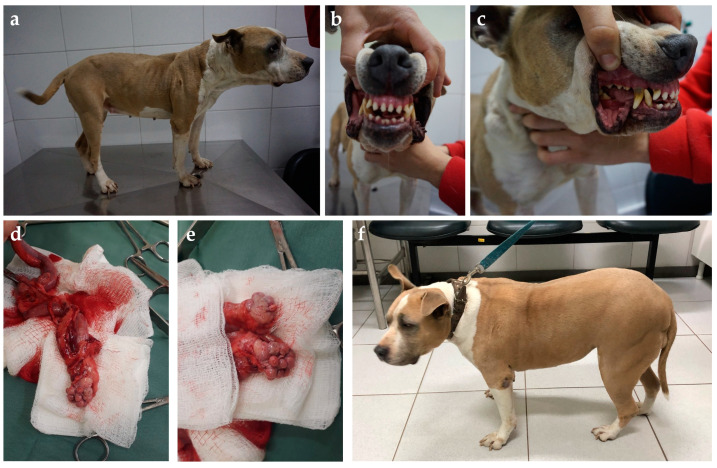
Five-year-old American Stratford Shire bitch presented with diabetes diagnosed weeks after the last heat. Acromegalic features such as (**a**) increased thick skin folds, (**b**) widened interdental space, and (**c**) discrete macroglossia and prognathism were observed. Panting and snoring were also reported by the owners. The bitch was submitted to ovariohysterectomy, and (**d**) an enlarged uterus filled with a serosanguinous content diagnosed as hematometra and (**e**) bilateral polycystic ovaries were found. (**f**) Weight gain and muscle mass were restored after adequate diabetic control, as well as skin folds becoming regressed; however, despite that diabetic remission was not achieved, glycemic control was consistently improved after surgery with a ~40% smaller insulin dose.

**Figure 7 animals-14-00890-f007:**
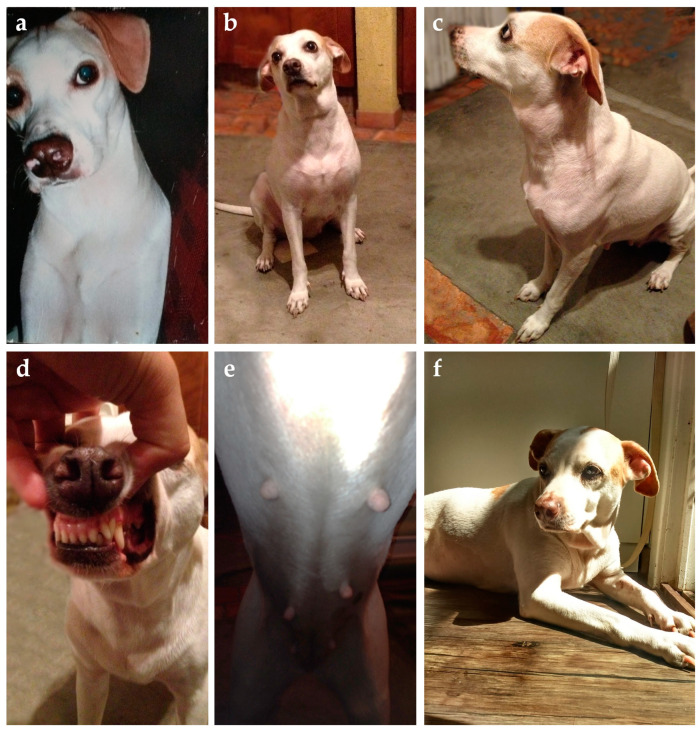
Evolution of acromegalic features over time in a mixed-breed intact dog. (**a**) Facial and neck skin aspect by the age of five years. By the age of eight years, there was (**b**,**c**) skin folding on the ventral neck and over the wither, and (**d**) increased interdental space was starting to become evident. (**e**) Mammary glands were well demarked despite being in anestrus and that mating never occurred. Ovariohysterectomy was performed as a preventive measure against PRDM when she was nine years old. (**f**) Increased skin folding in the ventral neck by the time surgery was performed. The bitch never developed diabetes and died two years later due to a liver tumor.

**Table 1 animals-14-00890-t001:** Dog breeds with a reported predisposition for pyometra and/or diabetes mellitus.

Increased Risk for Pyometra [12,13,110,111]	Increased Risk for Diabetes Mellitus [20,21,22,23,24,25,26]
Airedale Terrier	Australian Terrier
Bernese Mountain Dog	Bichon Frise
Bouvier de Flandres	Border Collie
Boxer	Border Terrier
Bull Terrier	Cairn Terrier
Bullmastiff	Cavalier King Charles Spaniel ^1^
Cavalier King Charles Spaniel ^1^	English Setter
Dogue de Bordeaux	English Springer Spaniel ^1^
Drever	Finish Spitz
English Cocker Spaniel	Fox Terrier
English Springer Spaniel ^1^	Irish Setter
German Shepherd Dog	Keeshond ^1^
Golden Retriever	Miniature Schnauzer ^1^
Great Dane	Poodle (toy/miniature) ^1^
Irish Terrier	Samoyed
Irish Wolfhound	Siberian Husky
Jämthund	Standard Schnauzer
Keeshond ^1^	Swedish Elkhound (female)
Labrador Retriever	Swedish Lapphund
Leonberger	Tibetan Terrier
Miniature Schnauzer^1^	Toy Poodle
Newfoundland	West Highland White Terrier ^1^
Poodle (toy/miniature) ^1^	Yorkshire Terrier
Rottweiler	
Rough Collie	
Saint Bernard	
Shetland Sheepdog	
Staffordshire Bull Terrier	
West Highland White Terrier ^1^	

^1^ Breeds with a described predisposition to both pyometra and canine diabetes mellitus.

**Table 2 animals-14-00890-t002:** Dog breeds with a reported decreased risk for pyometra and/or diabetes mellitus.

Decreased Risk for Pyometra [12,13,14,15,16,17,18,19,20,21,22,23,24,25,26,27,28,29,30,31,32,33,34,35,36,37,38,39,40,41,42,43,44,45,46,47,48,49,50,51,52,53,54,55,56,57,58,59,60,61,62,63,64,65,66,67,68,69,70,71,72,73,74,75,76,77,78,79,80,81,82,83,84,85,86,87,88,89,90,91,92,93,94,95,96,97,98,99,100,101,102,103,104,105,106,107,108,109,110]	Decreased Risk for Diabetes Mellitus [20,21,22,23,24,25,26]
Coton de Tulear	Airedale Terrier
Dachshund	Basset Hound
Finnish Spitz	Beagle
Gordon Setter	Boston Terrier
Laika	Boxer
Lancashire Bull Terrier	Brittany Spaniel
Maltese	Bulldog
Norrbotten Spitz	Cocker Spaniel
Norwich Terrier	Collie
Saluki	Dalmatian
Tibetan Terrier	Doberman Pinscher
	English Pointer
	English Setter
	German Sheperd Dog
	German Short-Hair Pointer
	Golden Retriever
	Great Dane
	Greyhound
	Irish Setter
	Norwegian Elkhound
	Old English Sheepdog
	Pekingese
	Shetland Sheepdog
	Shih Tzu

## Data Availability

No new data were created or analyzed in this study. Data sharing is not applicable to this article.

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
