# Peer review of "Progesterone-Related Diabetes Mellitus in the Bitch: Current Knowledge, the Role of Pyometra, and Relevance in Practice"

_animals, 2024, doi:10.3390/ani14060890_

Round 1

Reviewer 1 Report

Comments and Suggestions for Authors

Interesting, well organized, and comprehensively described work. The paper is scientifically valid and not misleading. The bibliography is appropriate. Far from providing definitive recommendations and understanding of characteristics related to PRDM in the bitch, this review bring out relevant insights and review the main aspects involved in this CDM subtype of particular interest. The complex effects of progesterone and progesterone-controlled growth hormone over expression during right-hand or pregnancy are discussed as risk factors for the development of DM. Sterilization is mentioned as a preventive measure against the development of the CDM, with the added benefit of drastically reducing the risk of promoter development. It is emphasized that early diagnosis of CDM and rapid surgical resolution of the pyrometer and termination of pregnancy are associated with better outcomes and a higher likelihood of diabetic remission.

Author Response

The Authors would like to Thank you for your time reviewing our manuscript and for the comments provided. 

Reviewer 2 Report

Comments and Suggestions for Authors

The presented manuscript is a nicely written review that focuses on the current knowledge concerning the relationship between estrus cycle (its luteal phase) and pyometra, as well as canine diabetes melitus. This topic is within the scope of the journal and interesting from both scientific and practical point of view. It should be stressed, that the above mentioned topic has been rather weakly explored and analyzed in scientific veterinary journals until now. Due to this, this topic is current and interesting for the readers, as well as there is still need to discuss and to analyze what has been done.

In my opinion, this manuscript is rather well-organized and relatively easy to follow and understand. The Authors performed an adequate review of the existing literature and additionally they enriched this text with a piece of own earlier experience on this field. Also the writing style of this paper is rather acceptable. Overall, it gives a nice summary of the current knowledge on the canine diabetes melitus with regard to the role of progesterone and inflammatory processes. Moreover, this article has been made more attractive thanks to some interesting Authors thesis for the future studies.

However, in my opinion there is still room for the editorial improvement of this manuscript. I would recommend to consider the following remarks:

·        the title of this paper perhaps should be better phrased, because not all items discussed are included,

·        in general, this review is overlong and wordy; due to this it would be profitable to condense the whole text – e.g. the subchapter 2 contains the very well known information (1 page!). Other particular remarks and adviceds are beyond this review.

In conclusion, this review is worth being published. I am aware, that my editorial remarks may be seem as subjective, that is why the final decision about their potential implementation should rest with Editor.

Author Response

The presented manuscript is a nicely written review that focuses on the current knowledge concerning the relationship between estrus cycle (its luteal phase) and pyometra, as well as canine diabetes melitus. This topic is within the scope of the journal and interesting from both scientific and practical point of view. It should be stressed, that the above mentioned topic has been rather weakly explored and analyzed in scientific veterinary journals until now. Due to this, this topic is current and interesting for the readers, as well as there is still need to discuss and to analyze what has been done.

Response: thank you for the comments and suggestion to stress how poorly covered the topic is in veterinary literature. The main focus of this review was exactly to bring up all the relevant information published in veterinary medicine and provide centralized relevant information for practitioners and Vet students interested in this topic. This objective was added to the manuscript emphasizing the topic was weakly explored in vet literature.

In my opinion, this manuscript is rather well-organized and relatively easy to follow and understand. The Authors performed an adequate review of the existing literature and additionally they enriched this text with a piece of own earlier experience on this field. Also the writing style of this paper is rather acceptable. Overall, it gives a nice summary of the current knowledge on the canine diabetes melitus with regard to the role of progesterone and inflammatory processes. Moreover, this article has been made more attractive thanks to some interesting Authors thesis for the future studies.

Response: we appreciate your comments, thank you.

However, in my opinion there is still room for the editorial improvement of this manuscript. I would recommend to consider the following remarks:

  • the title of this paper perhaps should be better phrased, because not all items discussed are included,

Response: we agree with this suggestion, and we thought it would be better just to mention as: “Progesterone-related diabetes mellitus in the bitch: current knowledge and why it is relevant?” Since pyometra and pregnancy are progesterone-related conditions, as well as it is synthetic progestogens use; we considered this new title more proper.

  • in general, this review is overlong and wordy; due to this it would be profitable to condense the whole text – e.g. the subchapter 2 contains the very well known information (1 page!). Other particular remarks and adviceds are beyond this review.

Response: we tried to summarize here the main aspects of the estrus cycle from a progesterone perspective to allow readers not familiar with hormonal fluctuations during the canine estrus cycle to be able to understand how the bitch reproductive endocrinology works.  We consider this physiology information crucial for adequate understanding and management of PRDM. In this way, we tried to shorten this section suppressing a few sentences. Also, progesterone was substituted by P4 over the text to reduce the total characters number.

In conclusion, this review is worth being published. I am aware, that my editorial remarks may be seem as subjective, that is why the final decision about their potential implementation should rest with Editor.

Response: we appreciate your opinion, and we would like to thank you for your time reviewing and helping to improve our manuscript.

Reviewer 3 Report

Comments and Suggestions for Authors

At first, the theme of the study, by providing a review of the interactions between diabetes and pyometra, is quite important and relevant. However, it must be recognized that over the years, numerous reviews on the topic have already been carried out. Therefore, it is important that this current review really brings relevant information, and a new approach, in order to differentiate itself from other existing ones, and contribute to a better understanding of the subject. Therefore, below, I list some aspects that need to be improved before a definitive evaluation:

1. The abstract is uninformative and generalist. I believe that there should be more focus on the differential aspects of the review, highlighting new information regarding the relationships between the estrous cycle, diabetes and pyometra, mainly with regard to etiologies and treatments. I believe that aspects of conservative treatments should also be expressed, and not just radical surgical treatment. Moreover, other aspects like diagnosis and etiology should be welcome here.

2. The introduction is extremely short and does little to emphasize the uniqueness of this review. Authors must make it clear why publishing a new review on pyometra would be worthwhile. In what aspects does this differ from the others? Highlight cases relating to the interaction between diabetes and pyometra, highlighting the importance of both diseases separately or in common.

2. The topic relating to the Canine Estrous Cycle does not provide any new information. It only compiles data that is already widely known from the perspective of dog reproduction specialists. If the authors' idea was to bring up the topic just to introduce the subject, perhaps it would be unnecessary to go into so much detail. In any case, this is just my point of view. The detail, obviously, is not wrong, just unnecessary given the context and focus of a new review.

3. On the topic regarding Diabetes Mellitus and Estrous Cycle, this is very well written. However, there is information on epidemiology and basic concepts that would better fit in the introduction, in order to emphasize the focus of the work and the innovative approach of this review. In truth, the manuscript should begin from these topic.

4. The topic regarding Progesterone - The Evil One is very informative. In fact, it already brings all the hormonal relationships involving diabetes and the estrous cycle. This makes it clearer that topic 2, regarding the estrous cycle, would be unnecessary.

5. The article is very textual. Graphs and tables would be welcome. For example, a table showing the relationships between dog breeds and diabetes would be very informative and would shorten the text.

6. Figure 1 is already very well known. Instead, authors could use organizational charts showing the relationships between, for example, progesterone, estrogen, growth hormone and insulin resistance. Such figures would be much more innovative than the classic graph of reproductive hormones in the canine estrous cycle. On the contrary, Figure 2 is really good. Other figures related to case reports are also welcome.

7. Reading the entire text, we can see the relevance of this review. It, in fact, brings an interesting approach to the subject and differs from other existing reviews. However, it is necessary for the authors to improve some parts of the work as mentioned above, in particular, the introduction and the abstract.

Author Response

REVIEWER 3 - Comments and Suggestions for Authors

At first, the theme of the study, by providing a review of the interactions between diabetes and pyometra, is quite important and relevant. However, it must be recognized that over the years, numerous reviews on the topic have already been carried out. Therefore, it is important that this current review really brings relevant information, and a new approach, in order to differentiate itself from other existing ones, and contribute to a better understanding of the subject. Therefore, below, I list some aspects that need to be improved before a definitive evaluation:

 Response: we have identified numerous reviews on pyometra, as well as reviews on the canine estrus cycle that were very helpful in building this review. Also, there are several general reviews on canine diabetes mellitus published in recent years. However, we have not identified any review focused on discussing the interactions among these topics in the universe of progesterone-related diabetes mellitus (PRDM). In this way, our aim was not to provide a deep review of the canine estrus cycle, pyometra, or canine diabetes mellitus, but to provide comprehensive material discussing the interactions among these topics, bringing in summarized information previously pulverized in veterinary literature. The reader interested in progesterone-related diabetes in the bitch will find deep material covering basic reproductive physiology, the impact of pyometra on insulin sensitivity, and insights about how to best manage and prevent PRDM.   

  1. The abstract is uninformative and generalist. I believe that there should be more focus on the differential aspects of the review, highlighting new information regarding the relationships between the estrous cycle, diabetes and pyometra, mainly with regard to etiologies and treatments. I believe that aspects of conservative treatments should also be expressed, and not just radical surgical treatment. Moreover, other aspects like diagnosis and etiology should be welcome here.

Response: thank you for your suggestions. The abstract was reviewed in light of your suggestions.

  1. The introduction is extremely short and does little to emphasize the uniqueness of this review. Authors must make it clear why publishing a new review on pyometra would be worthwhile. In what aspects does this differ from the others? Highlight cases relating to the interaction between diabetes and pyometra, highlighting the importance of both diseases separately or in common.

Response: Since the manuscript is a review article, we tried to keep the introduction as short as possible to avoid anticipating information that would be better discussed later. We improve the discussion following your suggestion of highlighting the importance of both diseases together or separately, as well as attending Reviewer 2’s suggestion of highlighting the importance of this review due to the paucity of papers in this field in the veterinary literature.

  1. The topic relating to the Canine Estrous Cycle does not provide any new information. It only compiles data that is already widely known from the perspective of dog reproduction specialists. If the authors' idea was to bring up the topic just to introduce the subject, perhaps it would be unnecessary to go into so much detail. In any case, this is just my point of view. The detail, obviously, is not wrong, just unnecessary given the context and focus of a new review.

Response: we tried to summarize here the main aspects of the estrus cycle from a progesterone perspective to allow readers not familiar with hormonal fluctuations during the canine estrus cycle to be able to understand how the bitch reproductive endocrinology works. At least in Brazil, we feel that many general practitioners have difficulties understanding hormone fluctuations during the canine estrus cycle.  We consider this physiology information crucial for adequate understanding and management of PRDM, and this justification was added in the end of the previous topic. We tried to shorten this section suppressing a few sentences. Also, progesterone was substituted by P4 over the text to reduce the total characters number.

  1. On the topic regarding Diabetes Mellitus and Estrous Cycle, this is very well written. However, there is information on epidemiology and basic concepts that would better fit in the introduction, in order to emphasize the focus of the work and the innovative approach of this review. In truth, the manuscript should begin from these topic.

Response: thank you for this suggestion. We moved the entire topic regarding diabetes definitions, epidemiology, and pathology to the beginning of the manuscript. We kept the shorter introduction and stressed that this topic is weakly explored in veterinary literature as suggested by Reviewer 2. In this way, topic 1 still is the introduction, and now topic 2 is named “The Relevance of Progesterone-Related Diabetes Mellitus Subtype”. The following topics were kept and renumbered. Topic 4 (previous topic 3) is now named just “4. Estrus Cycle Effects on Insulin Sensitivity” and immediately followed by the subtitle “4.1. Progesterone: The evil one?” and so on.

  1. The topic regarding Progesterone - The Evil One is very informative. In fact, it already brings all the hormonal relationships involving diabetes and the estrous cycle. This makes it clearer that topic 2, regarding the estrous cycle, would be unnecessary.

Response: we appreciate your opinion, thank you! Due to reasons exposed in the second topic of this reviewer's report, the section reviewing estrus cycle was kept and shortened as possible. Also, is important to note the physiological roles of progesterone are described in that section, while in the “Progesterone: The evil one?” information related to the influence of progesterone on insulin sensitivity is reviewed.

  1. The article is very textual. Graphs and tables would be welcome. For example, a table showing the relationships between dog breeds and diabetes would be very informative and would shorten the text.

Response: Thank you for this suggestion, we created two tables compiling breeds predisposed to pyometra and/or diabetes and breeds with reduced risk for pyometra/diabetes.

  1. Figure 1 is already very well known. Instead, authors could use organizational charts showing the relationships between, for example, progesterone, estrogen, growth hormone and insulin resistance. Such figures would be much more innovative than the classic graph of reproductive hormones in the canine estrous cycle. On the contrary, Figure 2 is really good. Other figures related to case reports are also welcome.

Response: Thank you for this suggestion. We created a new figure with a chart representing the main factors related to PRDM. Regarding Figure 1, since many veterinarians show difficulties in understanding the estrus cycle hormone fluctuations, we aimed to provide a colored graph emphasizing the main hormones involved bringing attention to the hormonal environment involved with increased risk for diabetes development. Another important contribution of this figure is the unique inclusion of the growth hormone profile during diestrus. Visually the reader can have an idea of how progesterone, prolactin, and GH behave during diestrus. This mental image is very welcome for “visual” people to understand the cyclic fluctuations.

  1. Reading the entire text, we can see the relevance of this review. It, in fact, brings an interesting approach to the subject and differs from other existing reviews. However, it is necessary for the authors to improve some parts of the work as mentioned above, in particular, the introduction and the abstract.

Response: Thank you for your time reviewing our manuscript and for all your suggestions to improve it.